# Evaluation of an Augmented Cognitive Behavioural Group Therapy for Perinatal Generalized Anxiety Disorder (GAD) during the COVID-19 Pandemic

**DOI:** 10.3390/jcm11010209

**Published:** 2021-12-31

**Authors:** Sheryl M. Green, Briar Inness, Melissa Furtado, Randi E. McCabe, Benicio N. Frey

**Affiliations:** 1Department of Psychiatry and Behavioural Neurosciences, McMaster University, Hamilton, ON L8N 3K7, Canada; rmccabe@stjoes.ca (R.E.M.); bfrey@stjoes.ca (B.N.F.); 2Women’s Health Concerns Clinic, St. Joseph’s Healthcare, Hamilton, ON L8N 3K7, Canada; innessbe@mcmaster.ca (B.I.); furtadom@mcmaster.ca (M.F.); 3Department of Psychology, Neuroscience and Behaviour, McMaster University, Hamilton, ON L8N 3K7, Canada; 4Anxiety Treatment and Research Clinic, St. Joseph’s Healthcare, Hamilton, ON L8N 3K7, Canada

**Keywords:** perinatal GAD, COVID-19 pandemic, Cognitive Behavioural Group Therapy, augmented treatment

## Abstract

The perinatal period is considered a window of vulnerability given the increased risk of psychiatric difficulties during this time, such as mood and anxiety disorders (ADs). Pre-pandemic rates of ADs in perinatal women were one in five but have since increased with the onset of the COVID-19 pandemic (COVID). In addition, recent research suggests that the focus of worry has shifted during the pandemic, with perinatal women reporting significantly more COVID-specific worries. The objective of this study was to augment our current evidence-based Cognitive Behavioural Group Therapy (CBGT) for perinatal anxiety protocol by targeting intolerance of uncertainty and tailoring existing strategies to address COVID-related worry and impact. Pregnant (*n* = 19) and postpartum (*n* = 49) women were recruited from regular clinic patient flow from a university-affiliated teaching hospital between September 2020 and March 2021. Improvements in generalized anxiety symptoms, worry, intolerance of uncertainty, and mood were observed at post-treatment, maintained at 3-months, and the intervention received high ratings of treatment satisfaction. This is the first study to examine an augmented CBGT for perinatal women with GAD during the pandemic and supports the inclusion of strategies that target intolerance of uncertainty as well as specific pandemic and perinatal worry content for effective outcomes.

## 1. Introduction

Pregnancy and the postpartum, often referred to as the perinatal period, are associated with increased risk of experiencing psychiatric difficulties such as mood and anxiety disorders [1,2]. Anxiety Disorders (ADs), in particular, affect up to one in five pregnant and postpartum women [3]. Since the onset of the COVID-19 pandemic (COVID), perinatal women appear to be at an even greater risk of experiencing significant anxiety, with 43–60% of perinatal women endorsing moderate to severe levels of generalized anxiety symptoms [4,5]. Given that perinatal ADs are associated with numerous adverse outcomes for both mothers and their infants [6,7,8,9,10], this increase in anxiety symptom severity is concerning.

Generalized Anxiety Disorder (GAD), characterized by excessive and difficult to control worry [11], is the most common AD in pregnancy and the postpartum period [3,12]. While people with GAD tend to endorse similar worry domains across the lifespan (e.g., worry about health or self and others, finances, work/school [13]), the focus of worry often shifts to reflect the current context and circumstance of one’s life [14,15,16]. The perinatal period is no exception, as pregnant and postpartum women with GAD tend to endorse worries that are predominantly maternally focused (e.g., parenting abilities, coping as a mother, the well-being of the infant [17], reflecting the current developmental context of their lives. Further, pandemic-specific anxiety has been documented in the general population (e.g., fear of infection [18]) and in perinatal samples (e.g., exposure risks for mother and baby; reduced social support; uncertainty of perinatal care [4,19]), suggesting that worry is also circumstance specific.

Given that the prevalence of perinatal anxiety has increased substantially since the onset of COVID, and the focus of worry often reflects the current context (e.g., perinatal) and circumstance (e.g., COVID) of life, our team qualitatively examined the worry content of 84 treatment seeking pregnant and postpartum women during the pandemic (between April 2020 to October 2020), the majority (94%) of which, had a diagnosis of GAD [20]. We found that 33.5% of participant’s principal worries were specific to COVID, and 40% of those COVID worries were specific to the perinatal context. Further, our results revealed that a significant number of COVID worries were related to reduced social support and uncertainty about the future. This is troublesome as both reduced social support and intolerance of uncertainty are associated with worsening of postpartum depression and anxiety symptoms [21,22,23].

Intolerance of uncertainty, defined as a dispositional characteristic that results from a set of negative beliefs about uncertainty and its implications and involves the tendency to react negatively on an emotional, cognitive, and behavioral level to uncertain situations and events [24], is associated with heightened distress and worry [25,26,27]. Individuals who are intolerant of uncertainty see ambiguity as stressful and anxiety provoking and believe that uncertain situations should be avoided. Further, they have the tendency to overestimate the possibility of unpredictable or negative events and make threatening interpretations of ambiguous information. Unsurprisingly, uncertainty has increased since the onset of the pandemic, as COVID represents an unprecedented challenge for everyone [28], but may be particularly challenging for those who are already intolerant of uncertainty. For instance, a recent study by Sbrilli and colleagues [29], found that elevated levels of intolerance of uncertainty in pregnant and postpartum women during the pandemic were associated with decreased self-reported mindfulness and increased psychological symptoms, such as depression and anxiety. Importantly, interventions that specifically include strategies targeting intolerance of uncertainty (e.g., behavioural experiments, mindfulness) have demonstrated reductions in worry and anxiety in non-perinatal populations [30,31,32]. As such, the need to target intolerance of uncertainty in perinatal treatment protocols is needed now more than ever and indeed the call to include them when adapting current protocols during the pandemic, has been made in recent studies [28].

Knowing that perinatal women with GAD are uniquely impacted by the COVID-19 pandemic provides justification for augmenting current treatment strategies to meet their unique mental health needs during this time. To date, no evidence-based psychological treatments have been developed for perinatal women with GAD that specifically target identified COVID-related worry content and its related impact despite the tremendous negative burden of COVID on pregnant and postpartum women [20]. As such, the objective of this study was to augment our current evidence-based Cognitive Behavioural Group Therapy (CBGT) for perinatal anxiety protocol [33,34], by including additional sessions that target intolerance of uncertainty, as well as tailoring existing cognitive and behavioural strategies to address COVID-related worry and impact identified by our qualitative research [20] and evaluate its effectiveness. We hypothesized that this augmented protocol would (1) significantly improve anxiety and worry, both in general and specifically related to COVID-19 specific worries, (2) reduce intolerance of uncertainty, (3) improve mood and (4) result in high treatment satisfaction by perinatal women with GAD.

## 2. Materials and Methods

### 2.1. Participants and Procedures

Participants were treatment-seeking patients of the Women’s Health Concerns Clinic (WHCC), St. Joseph’s Healthcare Hamilton, a university-affiliated teaching hospital and publicly funded outpatient clinic that specializes in the assessment and treatment of women’s mental health across the reproductive lifespan [35]. All procedures were approved by the Hamilton Integrated Research Ethics Board. Pregnant (*n* = 19) and postpartum (*n* = 49) women recruited from regular clinic patient flow, participated in this study. Pre-treatment assessments took place between September 2020 and March 2021. Eligibility criteria were as follows: (1) 18–45 years old; (2) pregnant or within the first 12 months postpartum; (3) a principal diagnosis of GAD confirmed through use of the Mini International Neuropsychiatric Interview for DSM-5 [36] and (4) fluent in English. Exclusion criteria were as follows: (1) active suicidal ideation and (2) current psychosis or substance use disorder. Assessments were completed by PhD-level clinical psychology graduate students with extensive training in semi-structured diagnostic assessments, and diagnoses were confirmed by a supervising licensed clinical psychologist. Participants also completed a demographic form (see Table 1 for participant sample details) along with a brief battery of questionnaires to further characterize the symptomatology of our sample. Self-report measures were re-administered within one to two weeks post-treatment, along with a treatment satisfaction questionnaire. In order to capture participants’ subjective interpretations of their improvements in COVID-related worry across treatment, participants were also asked to identify whether they were still experiencing excessive COVID worry and whether treatment effectively addressed their COVID worries at the post-treatment assessment. Finally, baseline questionnaires were readministered at three-months post-treatment to assess sustained symptom change. Each treatment group was led by a licensed clinical psychologist or senior PhD student in clinical psychology with extensive training in this protocol and co-led with a graduate-level clinical psychology trainee. Treatment groups were composed of 8 sessions, two hours in duration, occurring weekly. Importantly, at the time of data collection from pre-treatment to post-treatment (September 2020 to April 2021) vaccination in Canada had just started to become available as of January 2021. However, prioritization was to healthcare providers and front-line workers and was not readily available to others until later into the spring 2021. Further, the province of Ontario, Canada, where data collection was being conducted, went into lockdown in March 2021 to June 2021 due to the high numbers of COVID-19. As such, although vaccination rates are not known, participants in the study were likely unvaccinated (as availability was minimal to non-existent at the time of pre to post testing) and measures to control the virus (e.g., significant restrictions, lockdown) were in place, throughout the entire study.

#### Intervention-Augmented CBGT

The original CBGT treatment, based on our published manual [33] tested in a randomized clinical trial [34], was designed to target a range of anxiety symptoms as well as comorbid depressive symptoms. The traditional CBGT involved 6-weekly 2-h sessions in a small-group format (up to six participants per group; range = 4–6). Session content is tailored to address the unique challenges experienced by women with perinatal anxiety and depression (see [34] for details) with weekly assigned homework exercises designed to reinforce learning. The augmented CBGT protocol (Table 2) contained 2 additional sessions (for a total of 8). These additional sessions provided psychoeducation on COVID-19 (both general and specific perinatal facts), as well as cognitive and behavioural strategies tailored to focus on COVID-specific content. The cognitive and behavioural strategies in the traditional protocol were augmented to include critical COVID-related anxiety, worry, and impact content that was identified by pregnant and postpartum women with GAD in our previous study [20]. The two additional sessions (sessions 5 and 6 in the augmented CBGT) emphasized the role of intolerance of uncertainty in ADs. This content was informed by current CBT for intolerance of uncertainty protocols [37,38], but again, tailored to meet the unique needs of perinatal women and included behavioural experiments and mindfulness to increase one’s tolerance to uncertainty.

### 2.2. Measures

#### 2.2.1. GAD-7 (GAD-7) 

The Generalized Anxiety Disorder 7-Item Scale (GAD-7) [39] is a 7-item self-report questionnaire assessing severity of generalized anxiety symptoms experienced over the previous two-weeks [39]. Items on the GAD-7 are measured on a 4-point Likert scale ranging from 0 (not at all) to 3 (nearly every day). The GAD-7 has demonstrated good sensitivity (89%) and specificity (82%) in detecting a clinical diagnosis of GAD, when a cut-off score of 10 or higher is utilized [39]. The internal consistency of the GAD-7 in the current sample was α = 0.86.

#### 2.2.2. Penn State Worry Questionnaire (PSWQ)

The PSWQ [40] is a 16-item self-report measure that assesses worry and symptoms characteristic of GAD. Respondents are asked to rate each item on a 5-point scale ranging from ‘not typical at all’ (1) to ‘very typical’ (5), with total scores ranging from 16 to 80 and scores at or above 65 representing a clinically significant level of worry [41]. The PSWQ has demonstrated excellent internal reliability and validity across various populations [42], and has been utilized in both non-perinatal and perinatal samples [43,44]. The internal consistency of the PSWQ in the current sample was α = 0.78.

#### 2.2.3. Intolerance of Uncertainty Scale (IUS)

The IUS [45] is a 27-item self-administered questionnaire assessing one’s beliefs and reactions to uncertain events and ambiguity [45,46]. Items are scored on a 5-point Likert scale, ranging from 1 (not at all characteristic of me) to 5 (entirely characteristic of me), with total possible scores of 27 to 135. The IUS has demonstrated excellent internal consistency (α = 0.91–0.95) and good test–retest reliability (r = 0.78) in general (non-perinatal) populations. The IUS was recently validated for use as a screening tool for perinatal anxiety disorder risk, in which an optimal clinical cut-off score of 64 or greater was established [47]. The internal consistency of the IUS in this sample was α = 0.92.

#### 2.2.4. Edinburgh Postnatal Depression Scale (EPDS)

The EPDS [48] is a 10-item self-report questionnaire that assesses depression in both pregnant and postpartum women [49]. Items are scored on a 4-point Likert scale, ranging from 0 to 3, with higher scores indicating greater severity of depressive symptoms. The EPDS has demonstrated good sensitivity and specificity for a diagnosis of Major Depressive Disorder. Clinical cut-off scores of 15 or higher during pregnancy, and 13 or higher during the postpartum, are indicative of Major Depressive Disorder [50]. The internal consistency of the EPDS in this sample was α = 0.79.

#### 2.2.5. The Client Satisfaction Questionnaire (CSQ)

The CSQ [51] is a general self-report measure of satisfaction with health and human services, which can be used for a variety of settings. It elicits the client’s perspective on the value of services received. It consists of eight items that are to be answered on a 4-point Likert scale, and the overall score consists of item responses summed with a range from 8 to 32. Higher scores indicate higher satisfaction. This scale demonstrates good psychometric properties with high internal consistency (Cronbach’s alpha ranging from 0.92 to 0.93). In terms of construct validity, tests of global improvement correlated with client satisfaction (r = 0.53 [52]). The internal consistency of the CSQ in this sample was α = 0.89.

### 2.3. Statistical Analysis

Comparison of demographic and psychological variables, GAD-7, PSWQ, IUS and EPDS scores, between pregnant and postpartum participants, were assessed using independent *t*-tests (for continuous variables) and chi-square tests (for categorical variables). If continuous variables did not meet the assumptions of normality, as assessed by the Shapiro–Wilk test, the non-parametric Mann–Whitney U test was used. Repeated measures analysis of covariance assessed pre- to post-treatment and follow-up effects on primary (GAD-7) and secondary (PSWQ, IUS, EPDS) outcomes. No mathematical corrections were made for multiple comparisons. Effect sizes for all statistical tests were calculated. For *t*-tests, Cohen’s d values of 0.2, 0.5, and 0.8 are considered small, medium, and large, respectively. For chi-square tests and *z*-tests, phi and r values of ±0.1, 0.3, and 0.5, are considered small, medium, and large, respectively. All analyses were performed using IBM SPSS Statistics 23 statistical software [53].

## 3. Results

### 3.1. Participant Characteristics at Baseline

Participant demographics and characteristics are reported in Table 1. Pregnant participants ranged between 14 and 40 gestational weeks, with a mean of 30.06 (SD = 6.95) weeks. Postpartum participants ranged from 1–52 weeks postpartum, with a mean of 20.06 (SD = 11.82) weeks. There were no significant differences (*p* > 0.05) between pregnant and postpartum women with regard to age, ethnicity, marital status, education, comorbid mood or anxiety disorders, or psychotropic medication use. Understandably, postpartum women had significantly more children than pregnant women (*p* < 0.01).

There were no statistically significant or clinically meaningful differences in GAD (GAD-7), worry (PSWQ), or depression (EPDS) symptomatology between pregnant and postpartum women at baseline (*p* > 0.05), with both groups scoring above the clinical threshold on each of these measures. Notably, intolerance of uncertainty (IUS) was significantly higher in pregnant women (M_pre_ = 92.39) compared to postpartum women (M_pre_ = 79.91; t = 2.59, *p* = 0.01, d = 0.70). Given this distinction, pre- to post-treatment and follow-up analyses were conducted separately for pregnant and postpartum women.

### 3.2. Treatment Engagement and Discontinuation

In the present study, 68 (*n* = 19 pregnant and *n* = 49 postpartum) participants provided informed consent to participate. Fourteen participants (*n* = 4 pregnant and *n* = 10 postpartum) did not complete the augmented CBGT. Drop-outs were defined as those who completed less than seven treatment sessions or did not provide post-treatment data. Reasons for participant drop-outs were as follows: (1) loss of contact (*n* = 3 pregnant and *n* = 4 postpartum); (2) unable to make time commitment (*n* = 3 postpartum); (3) preferred individual services (*n* = 2 postpartum); (4) medical problem requiring attention (*n* = 1 pregnant); and (5) symptoms improved (*n* = 1 postpartum). The difference in participant drop-outs between pregnant and postpartum groups was not statistically significant [χ^2^(1) = 5.60, *p* = 0.23]. Of the 14 pregnant participants who completed group treatment, 10 transitioned to postpartum by the post-treatment assessment, and an additional 3 transitioned by the 3-month follow-up.

### 3.3. Pre to Post-Treatment

#### 3.3.1. Primary Outcome

All pre- to post-treatment outcomes for pregnant and postpartum women are presented in Table 3. Both pregnant and postpartum women demonstrated statistically significant and clinically meaningful reductions in GAD symptoms from pre- to post-treatment. There was no significant medication use by GAD symptom interactions from pre- to post-treatment in pregnant or postpartum women (all *p*-values > 0.05).

#### 3.3.2. Secondary Outcomes

With regard to mean clinical change in secondary symptom outcomes, there were no statically significant reductions in worry symptoms for pregnant women from pre- to post-treatment (*p* = 0.08). Pregnant women however, demonstrated clinically meaningful reductions in worry symptoms, as post-treatment PSWQ scores were below the clinical threshold for this measure. Pregnant women showed statistically significant and clinically meaningful reductions in depression (*p* = 0.001), and statistically significant reductions in intolerance of uncertainty symptoms (*p* = 0.010) from pre- to post-treatment. Postpartum women demonstrated statistically significant and clinically meaningful reductions in worry (*p* < 0.001) and depression (*p* < 0.001), as well as statistically significant reductions in intolerance of uncertainty symptoms (*p* < 0.001) from pre- to post-treatment. There was no significant medication use by worry, depression, or intolerance of uncertainty symptom interactions from pre- to post-treatment in pregnant or postpartum women (*p* > 0.05).

### 3.4. Post-Treatment to 3-Months Follow-Up

All symptom improvements made by pregnant and postpartum women during treatment were maintained from post-treatment to 3-months follow-up (see Table 4). Specifically, there were no significant differences in GAD symptoms (GAD-7: *p =* 1.00), worry (PSWQ: *p =* 0.13), depression (EPDS: *p =* 0.27), or intolerance of uncertainty (IUS: *p =* 0.31) for pregnant women, suggesting that treatment gains were maintained at 3-months follow-up. Similarly, there were no significant differences in GAD symptoms (GAD-7: *p =* 0.52), worry (PSWQ: *p =* 0.14), depression (EPDS: *p =* 0.43), or intolerance of uncertainty (IUS: *p =* 0.54) for postpartum women from post-treatment to 3-months follow-up. There were no significant medication use by GAD symptom, worry, depression, or intolerance of uncertainty interactions from post-treatment to 3-months follow-up in pregnant or postpartum women (all *p*-values > 0.05).

### 3.5. Treatment Satisfaction and Qualitative Outcomes

Treatment satisfaction was assessed at post-treatment and is reported for 14 pregnant women and 37 postpartum women. The majority of participants rated the treatment as ‘excellent’ (*n* = 7 pregnant, 50%; *n* = 23 postpartum, 62.2%) or ‘good’ (*n* = 7 pregnant, 50%; *n* = 13 postpartum 35.1%), reported that the treatment ‘helped’ (*n* = 8 pregnant, 57.1%; *n* = 6 postpartum, 16.2%) or ‘helped a great deal’ (*n* = 6 pregnant, 42.9%; *n* = 12 postpartum, 32.4%), and reported that they would recommend the treatment to others. The majority of pregnant (*n* = 13, 92.9%) and postpartum (*n* = 35, 94.6%) women reported being satisfied with the program.

Qualitative data collected at post-treatment also suggested that 71.7% of participants (*n* = 9 pregnant, 69.2%; *n* = 24 postpartum, 72.7%) were no longer experiencing excessive anxiety related to COVID-19 by the end of treatment. Similarly, 93.2% of participants (*n* = 12 pregnant, 100%; *n* = 29 postpartum, 90.6%) reported that the treatment addressed their worry content.

## 4. Discussion

Rates of generalized anxiety symptoms in perinatal women have increased substantially since the onset of COVID [54,55], with evidence indicating that the pandemic has uniquely affected the worry content and lives of perinatal women [4,5,19,20], resulting in reduced medical and social supports, as well as increased uncertainty. Given that no evidence-based psychological treatments have been developed that specifically target COVID-related worry and its impact, the objective of our study was to augment our current evidence-based CBGT for perinatal anxiety protocol [33,34] and evaluate its effectiveness in meeting the unique mental health needs of perinatal women with GAD during the pandemic.

There were no significant differences in baseline GAD, worry or depression symptomatology between pregnant and postpartum women, with both groups scoring above the clinical cut-offs on these respective measures. Notably, intolerance of uncertainty was significantly higher in pregnant women compared to postpartum women, warranting the separation of these groups for pre- to post-treatment and follow-up analyses. Given the already existing uncertainty during pregnancy and the associated risk with postpartum anxiety worsening [23], it is understandable as to why intolerance of uncertainty would be heightened in pregnant women versus postpartum women. Factors which may not have been as relevant for postpartum women during the pandemic, such as reductions in perinatal medical supports (e.g., decrease in frequency of appointments, partners unable to join medical appointments) may have acted as an additional contributor to increases in uncertainty beyond those seen in postpartum women.

Consistent with our hypotheses, augmented CBGT led to statistically significant and clinically meaningful reductions in GAD symptoms from pre- to post-treatment, with both pregnant and postpartum women scoring below the clinical threshold following treatment. Importantly, reductions in GAD symptoms were maintained at 3-months follow-up and these gains remained significant while controlling for psychotropic medication use. The magnitude of change in GAD symptoms from pre- to -post-treatment is comparable to other studies looking at the effectiveness of cognitive behavioural therapy for GAD in the general population [56,57,58] and in perinatal samples [20,34,59]. Further, our qualitative data demonstrating that nearly three quarters of participants reported no longer experiencing excessive anxiety related to COVID-19 by the end of treatment and nearly all participants reported that the augmented CBGT addressed their worry content is a particularly important study outcome. As described, worry content in GAD changes with context and circumstance and research has captured this phenomenon with perinatal women during the COVID-19 pandemic. The fact that nearly all participants reported that treatment addressed their worry content and that most participants no longer had excessive worry related to the pandemic, adds further validation to this protocol’s effectiveness.

Our augmented CBGT protocol also led to statistically significant and clinically meaningful reductions in secondary outcomes of intolerance of uncertainty and depression from pre- to post-treatment in pregnant women and clinically meaningful but not statistically significant reductions in worry. Postpartum women demonstrated statistically significant and clinically meaningful reductions in all secondary outcomes (i.e., worry, depression, intolerance of uncertainty) from pre- to post-treatment. Again, gains on all secondary outcomes were maintained at 3-months follow-up for pregnant and postpartum women and were not influenced by psychotropic medication use. Given that worry is a defining feature of GAD [11], and has been linked to maintenance of GAD symptoms [20,59,60], reduction of worry symptoms in our sample is promising.

With regard to depression symptomatology, depression is highly comorbid with GAD [61,62,63]. Further, depression occurring in pregnancy and the postpartum period has been linked to the development of adverse cognitive, social and emotional outcomes in offspring [64]. As such, effectively targeting depression during treatment for GAD is imperative. The fact that our protocol led to statistically and clinically meaningful reductions in depression symptoms for pregnant and postpartum women from pre- to post-treatment that were maintained at 3 months follow-up is encouraging, especially given pandemic-related physical distancing, lockdown, and social isolation restrictions issued by the government throughout this time.

Reductions in, and maintenance of intolerance of uncertainty symptoms in pregnant and postpartum women is another critical finding from our study, as intolerance of uncertainty in pregnancy has been linked to higher risk of anxiety worsening postnatally [23]. Further, emerging research suggests that more people are struggling with uncertainty during the pandemic [27], which has been associated with increased generalized anxiety, depression, and other mental health symptoms, as well as maladaptive coping strategies, both in the general population [27] and perinatally [28]. Although replication of our findings is necessary, our study provides support for the inclusion of strategies that specifically address intolerance of uncertainty in perinatal women, such as behavioural experiments and mindfulness, included in the present study.

The present study has several limitations. First, we did not use a comparison condition and our sample size was relatively small, which limits generalizability and interpretation of results. As such, future studies should consider evaluating augmented CBGT in a larger sample and against a waitlist-control or other active comparison conditions to further establish its effectiveness. In addition, given the small sample size and limited number of pregnant women in our sample by post-treatment and follow-up, we were unable to evaluate what effect gestational age or weeks postpartum had on treatment outcomes. Future studies should consider evaluating whether participating in treatment at various times throughout the perinatal period affects treatment effectiveness. Further, since we did not evaluate the effectiveness of our traditional 6-session CBGT protocol with perinatal women with GAD during the pandemic, we are not certain as to whether that protocol would have been less effective, as effective, or even more effective than our augmented CBGT protocol. In addition, although the present study evaluated maintenance of treatment gains over a three-month period, future studies would benefit from assessing the longer-term impact of the augmented CBGT on primary and secondary outcome measures. Given that the sociodemographic characteristics of our sample were relatively homogenous (i.e., Caucasian, married, highly educated), it is plausible that our augmented CBGT may not be effective for perinatal women of diverse sociodemographic backgrounds. Future studies should evaluate the effectiveness of our augmented CBGT in heterogenous samples of perinatal women. Finally, participants were allowed to use psychotropic medications during this study thus making our sample ecologically valid. While we found no medication use by symptom improvement interaction in our analyses, a future medication-free study could also be very informative.

## 5. Conclusions

To our knowledge, this was the first study to examine the effectiveness of an augmented CBGT for perinatal GAD that is tailored to address the unique mental health needs of pregnant and postpartum women during the COVID-19 pandemic. Our study provides empirical support for a brief, eight-session augmented CBGT protocol to address symptoms of GAD, worry, intolerance of uncertainty, and depression in pregnant and postpartum women irrespective of psychotropic medication use during the COVID-19 pandemic.

## Figures and Tables

**Table 1 jcm-11-00209-t001:** Participant Demographics and Characteristics.

	Pregnant (*n* = 19)	Postpartum (*n* = 49)	Difference
*** Age, M(SD)**	30.78 (4.88)	30.88 (4.33)	t(65) = −0.08, *p* = 0.94, d = 0.02
*** Number of children, M(SD)**	0.56 (0.86)	1.35 (0.56)	t(65) = −4.41, *p* < 0.01, d = 1.09
	***n* (%)**	***n* (%)**	
*** Ethnicity**			χ^2^(1) = 2.56, *p* = 0.77, φ = 0.20
**Caucasian**	16 (88.9)	43 (87.8)	
**African American**	0 (0)	1 (2)	
**Asian/Pacific Islander**	0 (0)	1 (2)	
**Hispanic**	0 (0)	1 (2)	
**Indigenous**	0 (0)	1 (2)	
**Other**	2 (11.1)	2 (4.1)	
*** Marital status**			χ^2^(1) = 0.07, *p* = 0.80, φ = 0.03
**Single**	1 (5.6)	2 (4.1)	
**Married/Common-Law**	17 (94.4)	47 (95.9)	
*** Highest education**			χ^2^(1) = 1.61, *p* = 0.95, φ = 0.16
**Some or completed high school**	1 (5.6)	3 (6.1)	
**Certificate/Professional Diploma**	10 (55.5)	23 (46.9)	
**Bachelor’s Degree**	4 (22.2)	15 (30.6)	
**Masters Degree**	3 (16.7)	7 (14.3)	
**Professional Degree (e.g., MD, JD)**	0 (0)	1 (2)	
**Comorbid anxiety disorders**	7 (36.8)	14 (28.6)	χ^2^(1) = 0.44, *p* = 0.51, φ = −0.08
**Comorbid mood disorders**	10 (52.6)	32 (65.3)	χ^2^(1) = 0.93, *p* = 0.34, φ = 0.12
**Current psychotropic medication use ****	5 (26.3)	24 (50)	χ^2^(1) = 3.53, *p* = 0.17, φ = 0.23

* 1 pregnant participant did not complete all demographic measures. ** 1 postpartum participant did not complete medication history questionnaire.

**Table 2 jcm-11-00209-t002:** Augmented 8-week CBGT protocol session by session content.

Session	CBGT Augmented Protocol Content
1	Introduction and Psychoeducation: Information about anxiety, perinatal anxiety, and impact of COVID-19; introduction to the cognitive-behavioral model, the role of thoughts in maintaining distress, common perinatal and COVID-19 related worries and impact, and symptom monitoring.
2	Identifying and Challenging Unhelpful Thinking: Identifying unhelpful thinking and thinking errors; introduction to three strategies for cognitive restructuring (i.e., Best Friend Technique, Evidence Technique, Possibility Pie) to generate more balanced thinking. COVID-19 examples utilized with each technique.
3	Helpful vs. Unhelpful Worry: Differentiating between productive and unproductive worry; introduction to a systematic approach to problem-solving for productive worry. COVID-19 examples utilized.
4	Targeting Problematic Behavior: Psychoeducation on the role that behavior can play in maintaining distress; identifying problematic behavior (e.g., excessive reassurance seeking, excessive checking, avoidance with COVID-19 content woven in); introduction to exposure-based behavioral experiments. Use of COVID-19 related examples to demonstrate strategy.
5 (new session)	Introduction to Myths and Facts of COVID-19 and Intolerance of Uncertainty: Both general and perinatal themed myths and facts offered along with resources for further facts. Introduction to intolerance of uncertainty and behavioural experiments to increase tolerance to uncertainty
6 (new session)	In-session Behavioural Experiments and Introduction to Mindfulness: Conducting behavioural experiments in session and an introduction to mindfulness within session practice as a means to target intolerance of uncertainty.
7	Managing Depression: Psychoeducation on depressive symptoms in the perinatal period, risk factors and prevalence, as well as the impact of COVID-19 on increased risk of depression such as withdrawing from activity and isolation; impact that hormones and other biological and psychosocial changes have on mood; introduction to behavioral activation and activity scheduling within the government restrictions and safety guidelines of the COVID-19 pandemic. Introduction to paced respiration.
8	Assertive Communication: Psychoeducation on assertive and other forms of communication (i.e., passive, aggressive, passive-aggressive) and their consequences; discussion of situations where assertive communication is particularly needed in the perinatal period and within the COVID-19 pandemic; strategies for increasing assertive communication (e.g., planning for a strategic approach; assertiveness script; broken record technique). Wrap-up and summary of learning; strategies for relapse prevention.

NOTE: A detailed description of session-by-session content of the CBGT protocol can be found in our published manual [33]. CBT: cognitive behavioral therapy; CBGT: cognitive behavioral group therapy.

**Table 3 jcm-11-00209-t003:** Pre- to Post-Treatment Symptom Change in Pregnant (*n* = 15) and Postpartum (*n* = 39) Participants.

	Pre-Treatment		Post-Treatment					
	Mean	SD	Mean	SD	F-Value	*p*-Value	η^2^_p_	Clinical Cut-Offs
**Pregnant**								
GAD-7 ^a^	11.64	4.89	8.93	4.29	5.01	0.045	0.295	10
PSWQ	68.87	7.75	62.20	9.53	9.79	0.08	0.430	65
EPDS	15.20	4.46	10.40	4.67	17.94	0.001	0.580	15
IUS ^a^	93.50	20.76	83.50	19.14	9.49	0.010	0.442	64
**Postpartum**								
GAD-7 ^b^	12.53	4.90	6.39	3.19	44.89	0.000	0.569	10
PSWQ	66.51	8.09	59.74	9.57	22.31	0.000	0.376	65
EPDS	14.03	3.63	10.13	3.50	23.56	0.000	0.389	13
IUS ^c^	82.14	15.68	72.41	14.45	15.50	0.000	0.307	-

^a^ 1 pregnant participant did not complete the GAD-7 and IUS at post-treatment; ^b^ 3 postpartum participants did not complete the GAD-7 at post-treatment; ^c^ 2 postpartum participants did not complete the IUS at post-treatment.

**Table 4 jcm-11-00209-t004:** Post-Treatment to Follow-up Symptom Change in Pregnant (*n* = 11) and Postpartum (*n* = 24) Participants.

	Post-Treatment		3-Month Follow-Up					
	Mean	SD	Mean	SD	F-Value	*p*-Value	η^2^p	Clinical Cut-Offs
**Pregnant**								
GAD-7 ^a^	7.56	3.21	7.22	2.73	0.000	1.000	0.000	10
PSWQ	59.09	8.81	59.27	8.67	2.74	0.133	0.233	65
EPDS	9.55	4.16	8.64	3.75	1.34	0.276	0.130	15
IUS ^a^	83.18	20.34	81.09	20.37	1.15	0.312	0.113	-
**Postpartum**								
GAD-7 ^b^	6.29	2.35	7.43	3.47	0.42	0.526	0.022	10
PSWQ ^c^	61.70	9.36	61.00	8.45	2.30	0.145	0.009	65
EPDS	9.70	2.57	10.75	4.29	0.64	0.432	0.028	13
IUS ^c^	75.33	14.86	76.86	19.29	0.38	0.543	0.020	-

^a^ 2 pregnant participants did not complete the GAD-7 and IUS at 3-month follow-up, ^b^ 3 postpartum participants did not complete the GAD-7 at 3-month follow-up; ^c^ 1 postpartum participant did not complete the PSWQ and IUS at 3-month follow-up.

## Data Availability

Data made available upon request due to restrictions (e.g., privacy or ethical).

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
