# Peer review of "Evaluation of an Augmented Cognitive Behavioural Group Therapy for Perinatal Generalized Anxiety Disorder (GAD) during the COVID-19 Pandemic"

_jcm, 2021, doi:10.3390/jcm11010209_

Round 1
Reviewer 1 Report
My review of the article entitled: Evaluation of an Augmented Cognitive Behavioral Group 2 Therapy for Perinatal Generalized Anxiety Disorder (GAD) 3 During the COVID-19 Pandemic Comments: no comparison to the group without intervention: we do not know whether the intervention worked, or maybe it worked because after that time the epidemic wave weakened, or the time since delivery has passed - some patients in the study took psychotropic drugs, and some did not - the authors did not describe statistical differences. The question is, were these drugs effective in the treatment group, or were the drugs not working since the scales were the same as in the non-drug group? - the size of the groups was small, only 54 people - it may be enough for research in patients with psychiatric disorders but it is still not much, statistical analysis in such cases is virtually impossible more than half of the cited publications were published over 10 years ago, current literature must be cited
Author Response
Reviewer 1:
My review of the article entitled: Evaluation of an Augmented Cognitive Behavioral Group 2 Therapy for Perinatal Generalized Anxiety Disorder (GAD) 3 During the COVID-19 Pandemic Comments:
1) no comparison to the group without intervention: we do not know whether the intervention worked, or maybe it worked because after that time the epidemic wave weakened, or the time since delivery has passed
- Unfortunately, our study did not include a comparison group (e.g., waitlist control, another active treatment comparison group). Although we found significant differences in GAD symptoms, mood symptoms, worry, and intolerance of uncertainty from pre to posttreatment that were maintained at three month follow up, we agree that it is a limitation to not have a control arm to confirm with certainty that the treatment outcome was related to the intervention and have acknowledged this in our manuscript. We can confirm, though, that at the time of data collection from pre-treatment to post-treatment (September 2020 to April 2021) vaccination in Canada had just started to become available as of January 2021. However, prioritization was to healthcare providers, front line workers, etc. and was not readily available to others until later into the spring 2021. Further, we can also offer that the province of Ontario, Canada, where data collection was being conducted, went into lockdown in March 2021 to June 2021 due to the high numbers of COVID-19. It is therefore accurate to state that people who participated throughout the study were not vaccinated (as availability was minimal to non-existent at the time of pre to post testing) and measures to control the virus (e.g., significant restrictions, lockdown) were in place, throughout. Information on COVID-19 restrictions and vaccination status throughout the time of data collection has also been included in the manuscript.
2) Some patients in the study took psychotropic drugs, and some did not - the authors did not describe statistical differences. The question is, were these drugs effective in the treatment group, or were the drugs not working since the scales were the same as in the non-drug group?
- Thank you for this comment. We have conducted statistical analyses to determine if there were any differences in baseline symptom scores (e.g., GAD-7, EPDS, etc.) between those participants who were and were not on psychotropic medications. No statistically significant differences in baseline scores on the GAD-7, PSWQ, and EPDS between these two groups were found (p>0.05), however, there was a statistically significant difference in baseline IUS scores (p=0.027). Given that our primary outcome measure was the GAD-7, and these baseline scores were not significantly different between participants who were taking psychotropic medications and those who were not, we determined that we could evaluate the effectiveness of the augmented CBT protocol in combining these groups, while controlling for baseline psychotropic medication use (as we did).
- Further, as is seen in other mental health populations, such as depression, many patients do not experience clinically significant symptom improvement despite psychotropic medication use. There, combined psychological interventions with psychotropic medications are often required.
3) - the size of the groups was small, only 54 people - it may be enough for research in patients with psychiatric disorders, but it is still not much, statistical analysis in such cases is virtually impossible more than half of the cited publications were published over 10 years ago, current literature must be cited
- Thank you for this comment. We have reviewed our references that were more than 10 years old and have updated some literature with more recent publications (6, 7, 8, 9, 31, 32, 57, 61). We have chosen to keep certain references that are more than 10 years old, as some refer to the measures used and psychometric properties of various scales for which more rigorous and updated publications are not available (39, 40, 41, 42, 45, 46, 48, 49, 50, 51, 52). Further, we have also chosen to keep certain references where the publications cited, despite their age, continue to reflect what is currently known and accepted about these topics (14, 15, 22, 24, 25, 26, 30, 62, 63).
With regard to sample size, we do appreciate that a larger sample size would be more advantageous. Many psychological treatment studies use similar sample sizes, especially in initial evaluations of treatment efficacy. Sample size of initial treatment trials typically include less than 50 participants per arm, with some studies recommending as few as 15-25 participants per arm with expected small to medium effect sizes (Whitehead et al., 2016). We have added this as a limitation to our paper and acknowledge that future studies should evaluate this protocol in a larger sample.
Reviewer 2 Report
Thank you for the opportunity to review this study of CBGT treatment on mental health outcomes among pregnant and postpartum women. The manuscript concluded strong positive effect of CBGT therapy which is very encouraging.
One suggestion is to include the statistics on the gestational age and the weeks of postpartum at baseline. Were there any women who transitioned from pregnancy to postpartum during the treatment duration, or during the 3-month follow-up period? How were they handled in the analysis?
How did the effect of the therapy compare between pregnant women to postpartum women, or did the effect vary depending on the gestational age or postpartum weeks?
There was no control arm in this study, and hence it is difficult to assess whether the effect was due to the treatment or other factors. For example, the covid vaccine became available during the study period. Did any study subjects receive the covid vaccine? Another contributing factor is the calendar time. Covid prevalence rates were different over different calendar time period. Also the Covid restrictions were more relaxed as time passed.
Finally, considering the number of outcomes in the study, please consider using multiple comparison adjustment.
Author Response
Thank you for the opportunity to review this study of CBGT treatment on mental health outcomes among pregnant and postpartum women. The manuscript concluded strong positive effect of CBGT therapy which is very encouraging.
1) One suggestion is to include the statistics on the gestational age and the weeks of postpartum at baseline.
- We have updated the manuscript to include these statistics, which can be found on page 7, under section 3.1
2) Were there any women who transitioned from pregnancy to postpartum during the treatment duration, or during the 3-month follow-up period? How were they handled in the analysis?
- There were 19 pregnant women at the baseline assessment, with 14 completing the CBT group and post-treatment questionnaires. Of these pregnant participants, 10 transitioned to postpartum by the time of the post-treatment assessment. An additional 3 transitioned to postpartum by the 3-month follow-up visit. Given that our sample size was not large enough to conduct multiple comparisons, we did not include this change from pregnancy to postpartum in our analyses. We have included this information in section 3.2.
3) How did the effect of the therapy compare between pregnant women to postpartum women, or did the effect vary depending on the gestational age or postpartum weeks?
- Thank you for this comment. Unfortunately, we were unable to evaluate what affect gestational age or weeks postpartum had on the treatment outcomes given our sample size. We acknowledge and agree that it would be very beneficial for future studies to evaluate whether participating in treatment at various times in the perinatal period affects treatment outcome and have included this in the limitations of the study, along with suggestions for future directions.
4) There was no control arm in this study, and hence it is difficult to assess whether the effect was due to the treatment or other factors. For example, the covid vaccine became available during the study period. Did any study subjects receive the covid vaccine? Another contributing factor is the calendar time. Covid prevalence rates were different over different calendar time period. Also the Covid restrictions were more relaxed as time passed.
- Thank you for this comment. We agree that these are important considerations that the other reviewer also pointed out.Unfortunately, our study did not include a comparison group (e.g., waitlist control, another active treatment comparison group). Although we found significant differences in GAD symptoms, mood symptoms, worry, and intolerance of uncertainty from pre to posttreatment that were maintained at three month follow up, we agree that it is a limitation to not have a control arm to confirm with certainty that the treatment outcome was related to the intervention and have acknowledged this in our manuscript. We can confirm, though, that at the time of data collection from pre-treatment to post-treatment (September 2020 to April 2021) vaccination in Canada had just started to become available as of January 2021. However, prioritization was to healthcare providers, front line workers, etc. and was not widely available to others until later into the spring 2021 (and even then, the roll out took time). Further, we can also offer that the province of Ontario, Canada, where data collection was being conducted, went into lockdown from March 2021 to June 2021 due to the high numbers of COVID-19. It is therefore accurate to state that people who participated throughout the study were not vaccinated (as availability was minimal to non-existent at the time of pre to post testing) and measures to control the virus (e.g., significant restrictions, lockdown) were in place, throughout. Information on COVID-19 restrictions and vaccination status throughout the time of data collection has also been included in the manuscript.
5) Finally, considering the number of outcomes in the study, please consider using multiple comparison adjustment.
- Thank you for this comment. Although correcting for multiple comparisons is suggested by some researchers and statisticians as, understandably, the false positive rate increases with additional comparisons, we ultimately decided not to correct for multiple comparisons. Others suggest not correcting for multiple comparisons as it inflates the false negative rate and confuses interpretation of results when data are not random but are true observations (Rothman, 1990). We have, however, made it explicit in the manuscript that we did not correct for multiple comparisons, so readers are informed while interpreting the results.
Round 2
Reviewer 1 Report
the manuscript has been corrected. Nevertheless, there are still some doubts regarding the study, but they result from the assumptions of the project and cannot be corrected at this stage.